

# Passive acoustic monitoring from profiling floats as a pathway to scalable autonomous observations of global surface wind

Louise Delaigue[1], Pierre Cauchy[2], Dorian Cazau[3], Julien Bonnel[4], Sara Pensieri[5], Roberto Bozzano[5], Anatole Gros-Martial[6], Christophe Schaeffer[7], Arnaud David[7], Paco Stil[1], Antoine Poteau[1], Catherine Schmechtig[8], Edouard Leymarie[1] and Hervé Claustre[1]

[1]Sorbonne Université, CNRS, Laboratoire d'Océanographie de Villefranche, LOV, 06230 Villefranche-sur-Mer, France

[2]Institut des sciences de la mer (ISMER), Université du Québec à Rimouski (UQAR), Rimouski, Québec, Canada

[3]ENSTA, Lab-STICC, UMR CNRS 6285, Brest, France

[4]Marine Physical Laboratory, Scripps Institution of Oceanography, University of California San Diego, La Jolla, CA, 92093, USA

[5]Institute for the Study of Anthropic Impact and Sustainability in the Marine Environment (IAS), Consiglio Nazionale delle Ricerche (CNR), Genoa, Italy

[6]Centre d'Études Biologiques de Chizé, CNRS, Villiers-en-bois, France

[7]NKE Instrumentation, Hennebont, France

[8]OSU Ecce Terra, UAR 3455, CNRS, Sorbonne Université, Paris Cedex, France

*Correspondence to*: Louise Delaigue (louise.delaigue@imev-mer.fr)

**Abstract.** Wind forcing plays a pivotal role in driving upper-ocean physical and biogeochemical processes, yet direct wind observations remain sparse in many regions of the global ocean. While passive acoustic techniques have been used to estimate wind speed from moored and mobile platforms, their application to profiling floats has been demonstrated only in limited cases and remains largely unexplored. Here, we report on the first deployment of a profiling float equipped with a passive acoustic sensor, aimed at detecting wind-driven surface signals from depth. The float was deployed in the northwestern Mediterranean Sea near the DYFAMED meteorological buoy from February to April 2025, operating at parking depths of 500–1000 m. We demonstrate that wind speed can be successfully retrieved from subsurface ambient noise using established acoustic algorithms, with float-derived estimates showing good agreement with collocated surface observations from the DYFAMED buoy. To evaluate the potential for broader application, we simulate a remote deployment scenario by refitting the acoustic model of Nystuen et al. (2015) using ERA5 reanalysis as a proxy for surface wind. Refitting the model to ERA5 data demonstrates that the float–acoustic–wind relationship is generalizable in moderate conditions, but high-wind regimes remain systematically biased—especially above 10 m s$^{-1}$. Finally, we apply a residual learning framework to correct these estimates using a limited subset of DYFAMED wind data, simulating conditions where only brief surface observations—such as those from a ship during float deployment—are available. The corrected wind time series achieved a 37% reduction in RMSE and improved the coefficient of determination ($R^2$) from 0.85 to 0.91, demonstrating the effectiveness of combining reanalysis with sparse in-situ fitting. This framework enables the retrieval of fine-scale wind variability not captured by reanalysis alone, supporting a scalable strategy for float-



based wind monitoring in data-sparse ocean regions—with important implications for quantifying air–sea
exchanges, improving biogeochemical flux estimates, and advancing global climate observations.
**1 Introduction**
Wind plays a fundamental role in driving ocean dynamics, air–sea fluxes of gases and
governing biological productivity and climate-related biogeochemical processes (Wanninkhof,
2014; McGillicuddy, 2016). Recent modelling studies emphasize that wind-driven ocean
circulation significantly influences regional climate trends, such as the North Atlantic Warming
Hole phenomenon (McMonigal et al., 2025). Despite its critical importance, accurately
quantifying oceanic wind variability remains challenging, particularly in remote and
undersampled regions such as the Southern Ocean, where satellite retrievals are limited by
coarse resolution, signal degradation from storms, heavy cloud cover, and sea ice (Bentamy et
al., 2003; Chelton et al., 2007; Verhoef et al., 2012). Consequently, observational gaps persist,
affecting our understanding of critical processes like air-sea carbon exchange during storm
events (Carranza et al., 2024).
Traditionally, oceanic wind observations have relied heavily on satellite scatterometry and
surface-based platforms, including meteorological buoys. While scatterometers provide near-
global wind observations, their effectiveness diminishes significantly under stormy conditions,
heavy precipitation, and seasonal ice coverage, limiting the accuracy and temporal resolution
required to capture highly dynamic atmospheric conditions at high latitudes (Chelton et al.,
2007; Verhoef et al., 2012). Surface platforms, although providing high-resolution data, suffer
from spatial limitations and high deployment and maintenance costs.
An alternative method with substantial promise involves using passive acoustic sensing of
underwater ambient noise generated by surface wind stress and wave-breaking activities. The
relationship between wind speed and high-frequency ambient noise (1–20 kHz) has been
extensively validated through theoretical and empirical studies (Vagle et al., 1990; Farmer et
al., 1998; Oguz and Prosperetti, 1990). These foundational studies demonstrated that air bubble
entrainment due to wave breaking, and raindrop impacts produces distinctive acoustic
signatures, offering a robust proxy for surface meteorological conditions. This approach builds
on the Weather Observations Through Ambient Noise (WOTAN) framework, formally
introduced by Vagle et al. (1990), which directly links wind-driven surface processes to
characteristic underwater acoustic signatures. The WOTAN methodology has since been
successfully implemented in dedicated instruments such as the Passive Acoustic Listener
(PAL), enabling autonomous and continuous monitoring of wind and rainfall from subsurface
acoustic recordings (Nystuen et al., 2001). Building upon this foundation, Ma et al. (2005)
developed a semi-empirical acoustic model capable of discriminating between wind-induced
and rain-induced ambient noise features, thereby enabling reliable estimation of wind speeds
from subsurface recordings. Subsequent studies extended these methods to drifting and



subsurface platforms, validating the acoustic–wind relationship across varied conditions (Ma
and Nystuen, 2005; Nystuen et al., 2015; Pensieri et al., 2015).
Advancements in passive acoustic sensing technology have enabled the integration of acoustic
sensors onto autonomous oceanographic platforms, including underwater gliders (Cazau et al.,
2018; Cauchy et al., 2018) and profiling floats equipped with PAL sensors (Riser et al., 2008;
Yang et al., 2015; Yang et al., 2016; Ma et al., 2023). Such developments are especially
valuable in remote environments, where traditional in-situ measurements remain limited. For
example, Menze et al. (2012) provided early evidence of wind-dependent acoustic noise
regimes in the Weddell Sea, while Cazau et al. (2017) and Gros-Martial et al. (2025) extended
these methods by using biologged southern elephant seals, demonstrating the feasibility of
estimating wind speed from passive acoustic recordings in the polar frontal zone. Beyond
atmospheric sensing, acoustic-equipped profiling floats have also proven valuable for a broader
range of geophysical and ecological applications, including detection and classification of
marine mammal vocalizations (Matsumoto et al., 2013; Baumgartner and Bonnel, 2022),
monitoring of hydroacoustic earthquake signals and ambient ocean noise (Pipatprathanporn
and Simons, 2022), and observing the presence of deep-diving cetaceans (Matsumoto et al.,
2013; Fregosi et al., 2020).
Despite these advancements, integration of passive acoustic sensors onto modern
biogeochemical (BGC)-Argo floats remains underexplored. BGC-Argo floats represent a
transformative technology in ocean observing, providing extensive datasets of critical oceanic
parameters including oxygen, nitrate, chlorophyll, pH, and downwelling irradiance (Johnson
and Claustre, 2016; Claustre et al., 2020). These autonomous platforms have significantly
improved our understanding of seasonal and interannual variability in nutrient dynamics
(Johnson et al., 2010), primary productivity (D'ortenzio et al., 2020), ocean acidification
(Williams et al., 2017), and carbon sequestration (Gray et al., 2018). Integrating acoustic wind-
sensing capabilities with BGC-Argo floats thus offers a unique opportunity to simultaneously
capture critical atmospheric forcing parameters alongside biogeochemical observations.
Recent technological developments, including miniaturized, low-power acoustic sensors
optimised for integration into autonomous platforms, now enable passive acoustic wind
estimation with minimal impact on float energy budgets and data transmission constraints
(Baumgartner et al., 2017). These advancements facilitate real-time onboard processing and
transmission of acoustic-derived environmental variables via satellite, thus overcoming
historical barriers associated with power consumption and data management. The integration
of acoustic sensors into BGC-Argo floats thereby holds promise for closing significant
observational gaps, particularly in undersampled regions such as the Southern Ocean.
Furthermore, the broader international scientific community has recognized the value of
passive acoustic sensing within global ocean observing frameworks. The Ocean Sound
Essential Ocean Variable (EOV), coordinated by the International Quiet Ocean Experiment



(IQOE) and endorsed by the Global Ocean Observing System (GOOS), specifically identifies
profiling floats as ideal platforms for scalable, distributed acoustic monitoring. This aligns with
current efforts to enhance autonomous ocean observing systems through multidisciplinary
sensor integration.
In this study, we present the first deployment of a profiling float equipped with a passive
acoustic sensor designed explicitly for wind speed estimation from subsurface ambient noise.
Deployed in the northwestern Mediterranean Sea, near the DYFAMED meteorological buoy,
this float serves as a proof-of-concept demonstration by integrating advanced acoustic sensing
with simultaneous biogeochemical measurements. Our main objective is to assess the
feasibility and precision of acoustic-based wind retrieval methods by applying and refining
established empirical algorithms tailored specifically to the acoustic characteristics of the
profiling platform. We validate float-derived wind estimates using collocated observations
from the DYFAMED buoy and the ERA5 atmospheric reanalysis dataset, highlighting both the
strengths and limitations of existing reference products. Finally, we propose a practical
framework whereby acoustic observations from the float can be effectively combined with
reanalysis data to enhance the accuracy of wind estimates in remote, data-sparse regions.
Through this approach, we demonstrate the potential of acoustic-equipped profiling floats to
serve as scalable, autonomous platforms within global ocean observing networks and capable
of closing critical observational gaps, improving quantification of air–sea exchanges, and
enriching our understanding of oceanic and climatic processes.



## 2 Materials and Methods

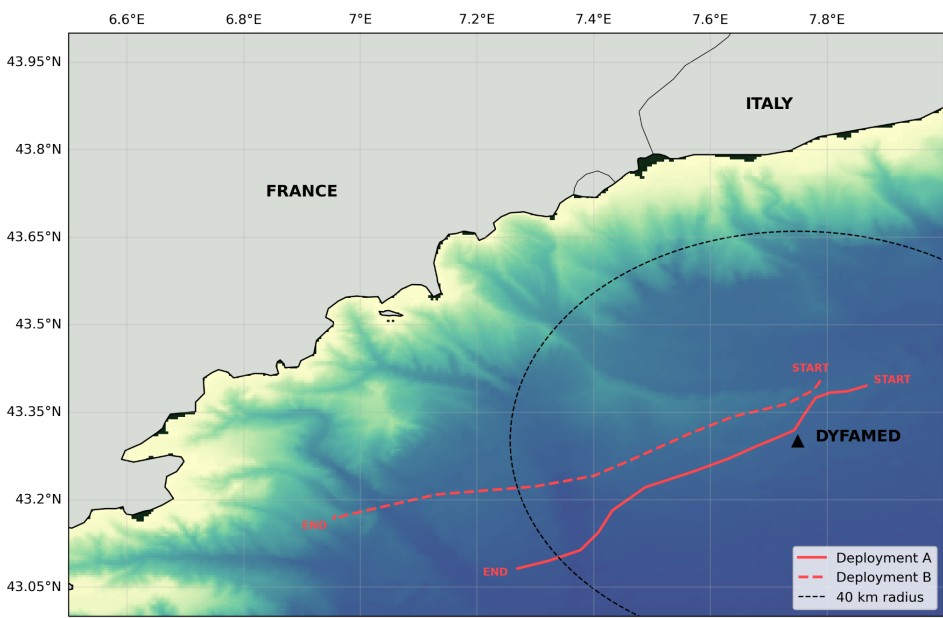

**Figure 1.** Float trajectories during sea trials conducted in the Ligurian Sea in February and March 2025. Deployment A (solid line) and Deployment B (dashed line) are shown along with a concentric dashed circle (40 km radius) centred on the DYFAMED station. The 40 km radius was used to spatially filter float data for refitting and validation of wind estimates at DYFAMED, as described in Cauchy et al. (2018).

### 2.1 Study area and DYFAMED weather station

The acoustic wind sensing trial was conducted in the Ligurian Sea, a sub-basin of the northwestern Mediterranean, in proximity to the DYFAMED (DYnamique des Flux Atmosphériques en MEDiterranée) oceanographic time series station (Fig. 1). DYFAMED (43.42°N, 7.87°E) has served as a key reference site for air–sea exchange, upper ocean dynamics, and biogeochemical cycling since the early 1990s. The station is equipped with continuous meteorological and oceanographic monitoring, including high-quality wind speed and direction measurements from a surface buoy maintained by Météo-France. These data are reported at hourly resolution, following WMO (World Meteorological Organization) standards, and include wind parameters, along with air temperature, pressure, humidity, and sea state. During the study period, wind speeds at DYFAMED ranged from 0.5 to 16.1 m s$^{-1}$, with a mean of 6.8 m s$^{-1}$ and a measurement precision of one decimal place.





## 2.2 Acoustic sensor integration

The float used in this study was equipped with a passive acoustic module jointly developed by NKE and ABYSsens in collaboration with LOV. This module was specifically designed for integration into the PROVOR CTS5 BGC-Argo platform, with the aim of minimizing power consumption and data volume while remaining compatible with the operational constraints of the BGC-Argo program.

The module consists of two main parts enclosed in a dedicated external housing: 1) a low-noise HTI-96-Min hydrophone (sensitivity: −165 dB re 1 V/µPa; frequency range: 2 Hz–30 kHz), mounted externally to capture pressure fluctuations, and 2) an ABYSsens acquisition board, which conditions, digitizes, and processes the signal.

The acquisition system operates in a low-power pulsed mode (220 mW) with a sampling frequency up to 62.5 kHz and 24-bit resolution. To limit power usage and transmission needs, raw acoustic signals are not stored. Instead, the sensor performs direct onboard integration into 23 third-octave bands, spanning from 63 Hz to 25 kHz with a variable integration time (see Table 1). Higher-frequency bands (e.g., 3.15–25 kHz) used shorter integration times (50 ms), while low-frequency bands used longer windows (up to 500 ms).

| Frequency band range | Integration time |
|:---:|:---:|
| **63**, 100, **125** and 160 Hz | 500 ms |
| **400**, 500 and 630 Hz | 250 ms |
| 800 Hz, **1**, 1.25, 1.6, **2** and 2.5 kHz | 100 ms |
| 3.15, 4, **5**, 6.3, **8**, 10, **12.5**, 16, **20** and 25 kHz | 50 ms |

**Table 1.** Integration times applied to third-octave bands during acoustic signal processing, varying by frequency range to balance energy and spectral accuracy. In bold and underlined, the bands transmitted in the "9 bands" float configuration.

The acoustic unit is mounted on the upper section of the float chassis and is configured to operate exclusively during the parking phase (500–1000 m depth). During this phase, the float drifts with only routine background measurements (e.g., pressure, CTD), and acoustic acquisition is automatically suspended whenever noisy operations such as ballast pumping or CTD sampling occur, thereby avoiding contamination from self-noise.

The float system allows for flexible and modifiable configuration via satellite: the user can define the number of bands transmitted (23, 9, or a compact onboard estimate of wind/rain),





the acquisition interval (typically 5–15 minutes), and the number of acoustic samples averaged
per measurement. In this study, we used a 5-minute interval with 10 averaged acquisitions per
measurement (each acquisition is a spectral estimation using the integration times defined in
Table 1).
The telemetry and energy impact of adding an acoustic sensor to a 6-variable biogeochemical
float was evaluated by using the programming interface provided by NKE. The estimated
reduction in the number of cycles varies from 18% for acquisition every 5 minutes to 7% for
acquisition every 15 minutes during the whole parking drift of a 10-day Argo cycle and with 5
averaged acquisitions per acoustic measurement. The data volume increase depends on the
transmission format: from ~9% for onboard wind–rain estimates (15-min period) to ~85% for
a full 23-band spectrum (5-min period). A 9-band spectrum every 15 minutes—a likely
recommended setup—adds ~16%. These overheads remain within the platform's capacity,
confirming compatibility with concurrent BGC measurements.
Each sensor output transmitted by the float corresponds to the Third Octave Level (TOL), i.e.,
the sound pressure level integrated over a third-octave band, expressed in dB re 1 μPa. These
TOLs represent the float's primary spectral product and are used as input to the wind speed
retrieval models. The amplitude resolution of the transmitted data is 0.2 or 0.5 dB, with a
dynamic range up to 127 dB. This discretisation arises because the data are transmitted as
integers to save bandwidth, which requires selecting a resolution step.

### 2.3 Depth correction and spectral normalization

To account for the attenuation of surface-generated noise with depth, a correction was applied
to all acoustic measurements (Fig. 2). In this study, the correction term was calculated from the
first temperature–salinity profile (Fig. 2a-b) and applied throughout the deployment, as the
float remained in relatively stable hydrographic conditions (Fig. 2c). For long-term or basin-
scale missions, however, this coefficient would need to be recomputed for each profile, since
temperature and salinity variability along the float trajectory can significantly affect sound
propagation.
Following Cauchy et al. (2018), the correction takes the form:

$$\text{TOL}_0(f) \ = \ \text{TOL}(h, f) \ + \ \beta(h, f) \tag{1a,}$$

$$\text{where } \beta(h, f) = \ -10 \log \left\{ 2 \int_0^\infty \left[ \frac{r \, sin^2 \theta_{r,h} \, e^{-\alpha_f \, l_{r,h}}}{l_{r,h}^2} \right] dr \right\} \tag{1b,}$$





with TOL$(h, f)$ as the raw TOL measurement from the profiling float, h as the sensor depth, f
the centre frequency of the band, r the horizontal distance from a surface noise source to the
point vertically above the sensor, l the total pathlength between source and receiver (accounting
for depth and refraction), including refraction effects, θ the angle between the emitted acoustic
ray and the horizontal axis, and α the frequency-dependent attenuation coefficient for bubble-
free water. The integral considers contributions from all surface-generated acoustic sources
over the sea surface, assuming radial symmetry, and accounts for geometric spreading,
frequency-dependent absorption, and angle-dependent energy emission along each path. This
correction was originally derived for third-octave levels and is directly applicable here, as the
float outputs TOLs at fixed centre frequencies.
Then, depth-corrected third-octave levels (in dB re 1 µPa) were converted to spectral density
levels (dB re 1 µPa/Hz) by normalising to the bandwidth of each band. This step ensures
consistency across frequencies and comparability with model spectra. In future deployments,
this spectral correction will be applied directly onboard the float.

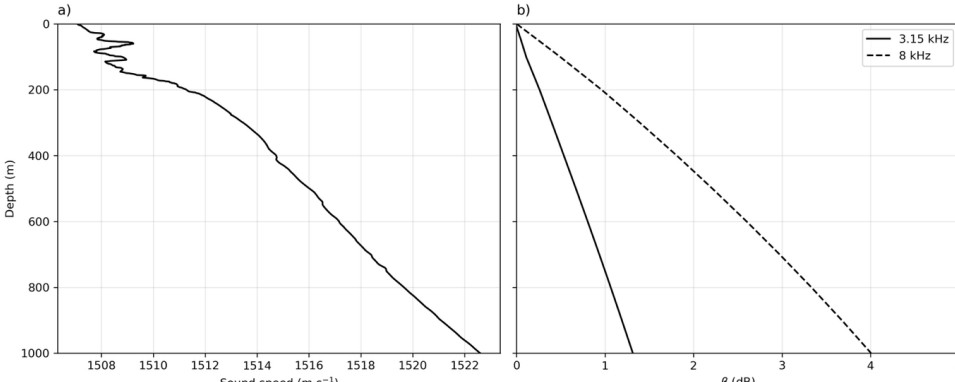


**Figure 2**. a) Sound speed profile used to derive the b) depth correction term $\beta(h, f)$ as a function
of depth, following the formulation of Cauchy et al. (2018). The correction accounts for the
attenuation of wind-generated surface noise with increasing sensor depth and was applied prior
to wind speed estimation. Here, β is shown at 3.15 kHz and 8 kHz.



## 2.4 Profiling float deployments

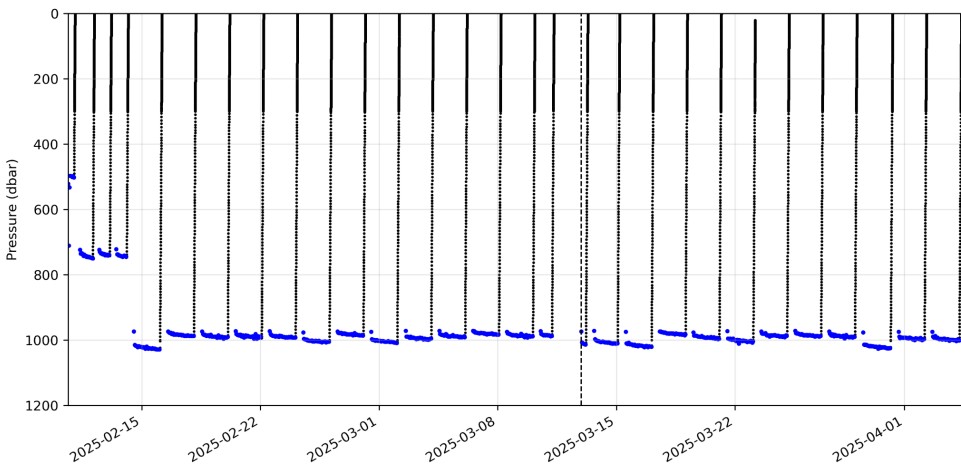


**Figure 3**. Vertical profiles from the acoustic-equipped profiling float deployed near
DYFAMED between February and April 2025. Blue points indicate times when passive
acoustic data were successfully recorded. The vertical dashed line marks the transition between
Deployment A and Deployment B.

Two deployments of an acoustic-equipped float (PROVOR CTS5) were carried out near
DYFAMED between February and April 2025 (Fig. 1). Deployment A lasted 30 days, from 10
February to 11 March, and Deployment B continued for 24 days starting on 12 March and
remained active until 4 April. The float operated in park-and-profile mode at three parking
depths (500, 700, and 1000 m; Fig. 2), collecting biogeochemical data during ascent and
passive acoustic data exclusively during the parking phases to minimize self-generated noise.
While Riser et al. (2008) previously demonstrated the feasibility of acoustic wind sensing from
Argo floats, their system transmitted only pre-processed wind estimates derived onboard using
a simplified version of the algorithm by Nystuen et al. (2015), without retaining or transmitting
spectral band data. This limited the possibility of reanalysis or applying alternative processing
schemes. In contrast, the floats used in this study recorded and transmitted full third-octave
band spectra, enabling detailed post-processing and algorithm refinement tailored to the float's
specific acoustic characteristics.
## 2.5 Transient and anthropogenic noise mitigation
Transient noise (i.e. episodic non-wind-related events) was mitigated by removing values
exceeding the 99th percentile within a ±1.5-hour window centred around each matched
timestamp. While this approach risks excluding some high-wind events, we verified that



extreme wind episodes typically span durations longer than a few hours, minimizing the chance
of misclassification (see Fig. 8).
To further reduce short-term variability and emphasize quasi-stationary wind-driven acoustic
patterns, we applied a 3-hour rolling mean to each frequency band. This choice reflects a
compromise between noise reduction and temporal resolution: the smoothing is sufficient to
stabilize wind estimates in the presence of submesoscale variability and intermittent noise, yet
long enough to preserve multi-hour wind events of interest. While this approach may attenuate
very brief fluctuations, our inspection of the time series suggests that the smoothing is sufficient
to suppress noise while retaining multi-hour processes of interest (eg., air–sea fluxes).
Alternative strategies, such as post-processing the wind speed estimates rather than the spectral
bands, could be explored in future deployments if finer-scale variability is a priority.
To mitigate anthropogenic noise contamination, Automatic Identification System (AIS) ship
tracking data were used to identify vessel presence within a 10 km radius and ±30 minutes of
each float timestamp. Acoustic observations were flagged as potentially contaminated if they
coincided with ship presence *and* showed anomalous deviations—defined as float-derived
wind speed differing from the DYFAMED buoy estimate by more than the root mean square
error (RMSE) observed under uncontaminated conditions. While this introduces a partial
dependence on external wind reference data, the combined AIS+anomaly criterion reduces
false positives and avoids relying solely on model–sensor differences for data exclusion. Data
flagged as contaminated were excluded from further analysis.



**2.6 Application of established acoustic models**

| Model | Input units | Wind frequency band (kHz) | Wind retrieval frequency (kHz) |
|---|---|---|---|
| Vagle et al. (1990) | dB re 1 µPa²/Hz | 7.1–8.9 | 8 |
| Nystuen et al. (2015) | dB re 1 µPa²/Hz | 7.1–8.9 | 8 |
| Pensieri et al. (2015) | dB re 1 µPa²/Hz | 7.1–8.9 | 8 |
| Cauchy et al. (2018) | dB re 1 µPa | 2.8–3.55 | 3.15 |


**Table 2.** Summary of acoustic wind speed estimation models and their input requirements.
Input units refer to the spectral level units used in model calibration. Central frequency
indicates the nominal retrieval frequency, and the third-octave band column specifies the
corresponding bandwidth. All models were calibrated and validated against standard 10-m
wind speed.
Empirical models have long been used to estimate surface wind speed from underwater ambient
noise, exploiting the link between wind-driven bubble formation and acoustic energy in the 1–
20 kHz band. These models typically relate surface wind speed U to the sound pressure level
$L_f$ measured in selected frequency bands. While many models use third-octave bands, others
rely on custom-defined or narrowband frequencies, often with variable bandwidths (e.g., 16%
of the centre frequency in Vagle et al., 1990).
We applied four established wind retrieval models spanning a range of functional forms—
cubic, two-regime linear–quadratic, composite, and two-regime log–linear. All wind models
were applied using acoustic levels consistent with their original formulations (Table 2). This
diversity allowed us to assess sensitivity to model structure and evaluate performance under
float-specific conditions. Each model was first implemented using its published coefficients to
generate wind speed estimates from float acoustic data, and the results were evaluated against
collocated meteorological observations (Fig. 4). Subsequently, the parameters of each model
were refitted using collocated float acoustic and wind data from the DYFAMED
meteorological buoy (Figs. 4 and 5; see Table 1 in Supplementary Material), which provides



hourly 10-meter wind speed. Model refitting was performed using nonlinear least-squares
optimization (Table 3). Wind records from DYFAMED were matched to float measurements
by nearest timestamp.
Following the spatial filtering approach of Cauchy et al. (2018), only float data within 40 km
of DYFAMED were retained for refitting and validation (Fig. 1). This threshold corresponds
to the estimated confidence radius around the DYFAMED meteorological buoy, within which
wind speed measurements show high spatial coherence (R = 0.86, RMSE = 2.5 m s$^{-1}$) when
compared to the AROME-WMED atmospheric model (Rainaud et al., 2016). The updated
coefficients were then used to generate wind estimates over the full float dataset. While this
spatial proximity improves wind representativeness, it does not account for variations in wind
fetch, a parameter known to influence ambient noise generation, particularly through wave and
bubble field development (e.g., Prawirasasra et al., 2024).
These four models were selected to represent a range of analytical formulations commonly
used in acoustic wind retrievals. They all use frequency bands where wind-driven bubble noise
typically dominates the local ambient sound field, with reduced interference from low-
frequency sources such as distant shipping. Our aim was not to exhaust all available models,
but rather to evaluate a representative subset under consistent float-specific conditions,
emphasizing the effect of model structure and local fitting.
The specifications and key features of each model are summarized in Table 2 for reference.
For all models and validation steps throughout the rest of Methods section, wind speed refers
to the standard 10-meter wind speed, consistent with both the ERA5 reanalysis product and the
DYFAMED buoy observations used for calibration and evaluation.
The first model, from Vagle et al. (1990), was derived from moored hydrophone data in the
North Atlantic and relates wind speed to high-frequency noise at 8 kHz using a cubic
formulation:

$$U_{\text{Vagle 1990}} = 10^{\left(\frac{-38.70 + \sqrt{-38.70^2 - 4.7 \cdot 38 \cdot (\text{SPL}_{\text{8kHz}} - 21.69)}}{-7.38 \cdot 2}\right)} \qquad (2).$$

Next, we applied the cubic model from Nystuen et al. (2015), developed using long-term
acoustic records from fixed hydrophones in both the Pacific and Atlantic. This model targets
wind-generated noise at 8 kHz and includes band-specific criteria to distinguish wind
contributions from other sources such as rain and shipping (Table 2).

$$U_{\text{Nystuen 2015}} = 0.0005 \cdot \text{SPL}_{\text{8kHz}}^3 - 0.0310 \cdot \text{SPL}_{\text{8kHz}}^2 + 0.4904 \cdot \text{SPL}_{\text{8kHz}} \\ + 2.0871 \qquad (3).$$



We then tested the two-regime linear–quadratic model from Pensieri et al. (2015) at 8 kHz,
developed using moored hydrophone data from the Ligurian Sea, near our study area.
Calibrated for Mediterranean conditions, the model relates wind speed to ambient noise levels
at the 8 kHz band, applying distinct linear and quadratic fits across low- and high-noise
regimes. Notably, the transition between regimes is defined at 38 dB, corresponding to a wind
speed of 2.39 m s$^{-1}$ in their framework. However, it is important to note that the threshold
separating high and low regimes is not standardized across the literature and may vary between
studies.

$$U_{\text{Pensieri 2015}} = \begin{cases} 0.044642 \cdot \text{SPL}_{8\text{kHz}}^2 - 3.2917 \cdot \text{SPL}_{8\text{kHz}} + 63.016 \\ 0.1458 \cdot \text{SPL}_{8\text{kHz}} - 3.146, \text{for SPL}_{8\text{kHz}} < 38 \text{ dB} \end{cases} \tag{4}.$$

Finally, we included the two-regime log–linear model from Cauchy et al. (2018), developed
using acoustic data from a glider operating in the western Mediterranean. Designed for mobile
platforms, the model relates wind speed to third-octave noise levels centred at 3 kHz. The
model uses distinct logarithmic and linear fits across two noise regimes.
This choice of 3 kHz, instead of the more commonly used 8 kHz, was based on empirical
observations showing greater dynamic range and lower variance in this band, which may reflect
sensor-specific factors or the sensor's mounting configuration on the glider (Cauchy et al.,
2018). The relationship goes as:

$$U_{\text{Cauchy 2018}} = \begin{cases} \dfrac{1}{0.4 \cdot 10^4} \cdot \left( 10^{\frac{\text{SPL}_{3\text{kHz}} - S_{\text{off}}}{20}} + 0.2 \cdot 10^4 \right) \\ \dfrac{1}{1.6 \cdot 10^4} \cdot \left( 10^{\frac{\text{SPL}_{3\text{kHz}} - S_{\text{off}}}{20}} + 12.5 \cdot 10^4 \right) \text{for } U > 10 \text{ m s}^{-1} \end{cases} \tag{5}.$$

The wind retrieval relationship is modelled using a two-regime log-linear function. The
transition between regimes occurs at wind speeds of approximately 10–11 m s$^{-1}$, established
empirically. To represent this switching behaviour, a relative threshold level is introduced,
expressed as SPL − S$_{\text{off}}$, where S$_{\text{off}}$ denotes the sea-state 0 noise reference. This formulation
highlights when wind-driven noise becomes dominant relative to the reference background
noise.
**2.7 Simulated wind estimation using reanalysis and residual learning**
To evaluate the ability of float-derived acoustic measurements to estimate surface wind speed
in regions without direct atmospheric observations, we used wind data from the ERA5
atmospheric reanalysis produced by the European Centre for Medium-Range Weather



Forecasts (ECMWF; Bell et al., 2021). ERA5 provides global wind fields on a 0.25° × 0.25°
spatial grid with hourly temporal resolution, offering a consistent and widely used reference
for surface atmospheric conditions.
Hourly ERA5 data were retrieved for the period spanning the float deployments, from 10
February to 31 March 2025. Specifically, we extracted the 10 m zonal ($u_{10}^2$) and meridional
($v_{10}^2$) wind components from the grid cell containing the float's position. Wind speed (U) was
then computed as:

$$U = \sqrt{u_{10}^2 + v_{10}^2} \tag{6}.$$

These values were time-matched to float and DYFAMED measurements using the nearest
available ERA5 hour.
Using ERA5 wind speeds as a reference, we refitted the empirical model from Nystuen et al.
(2015; 3) to float-measured Sound Pressure Level (SPL) at 8 kHz, producing a new set of
coefficients tailored to the float deployment. This produced a first-pass wind estimate derived
from float acoustics alone, calibrated to ERA5 rather than to DYFAMED in-situ observations.
This approach simulates a scenario in which a profiling float is deployed in a remote region
lacking surface wind measurements, and reanalysis products are used to train or tune the
acoustic model.
To improve the accuracy of this ERA5-calibrated estimate, we developed a residual learning
framework that uses limited collocated DYFAMED in-situ observations to correct systematic
errors. This training set, consisting of observations within 40 km, represents approximately
40% of the full dataset. This setup was designed to simulate a realistic scenario where ship-
based wind measurements are available in proximity to a float deployment. Specifically, we
used wind speed measurements from the DYFAMED buoy to model residual differences
between the ERA5-based acoustic prediction and true surface conditions. A feature matrix was
constructed including SPL at 8 kHz, ERA5 wind speed (10-meter), normalized time
(deployment day), and the acoustic model prediction wind speed from Nystuen et al. (2015;
Eq. 3). Residuals relative to DYFAMED wind speed were modelled using XGBoost regression,
a gradient boosting machine learning algorithm based on gradient-boosted decision trees and
known for its high predictive performance and ability to handle non-linear relationships and
interactions between features (Chen and Guestrin, 2016).
To estimate prediction uncertainty, we applied bootstrapping at two levels. For the ERA5-
calibrated acoustic estimate, we generated 100 bootstrap samples by resampling the float
dataset with replacement and perturbing the ERA5 wind input using its reported uncertainty
(standard deviation σ = 1.5 m s⁻¹; Bell et al., 2021). The empirical model was re-fitted for each
bootstrap, and the resulting ensemble of predictions was used to compute the standard deviation



at each time point. This approach captures both the impact of ERA5 input uncertainty and
variability in the fitted model parameters.
For the ML-corrected wind speed, we trained an ensemble of 100 XGBoost models on
bootstrapped subsets of the training data. During both training and prediction, Gaussian noise
(mean = 0, $\sigma$ = 1.5 m s$^{-1}$) was added to the ERA5 wind feature to simulate observational
uncertainty. The Gaussian assumption provides a tractable way to propagate uncertainty
through the learning framework and is commonly used in ensemble perturbation methods when
only first- and second-moment statistics are available. While the true distribution of ERA5
errors may deviate from normality, the central limit tendency of aggregated atmospheric errors
makes the Gaussian approximation a reasonable first-order choice. Importantly, this approach
ensures that the output uncertainty reflects both the variability of the fitted ML model and the
stated input uncertainty, though future work could refine the noise model if detailed error
distributions become available. Final corrected wind speeds were computed by summing the
Nystuen et al. (2015) ensemble-mean prediction with the ensemble-mean residual. Uncertainty
bounds were defined as ±1$\sigma$, combining variability across the XGBoost ensemble with ERA5
input uncertainty in quadrature. Uncertainty for the ML-corrected estimate reflects the
variability of the residual model and ERA5 input uncertainty but does not propagate the
bootstrap spread of the underlying Nystuen fit, which we report separately.
This method demonstrates how passive acoustic observations from profiling floats can be
combined with global reanalysis products and limited in-situ data to improve local wind speed
estimates, simulating the upscaling of BGC-Argo float deployments in remote ocean regions
lacking direct wind speed estimates.

## 3 Results and Discussion

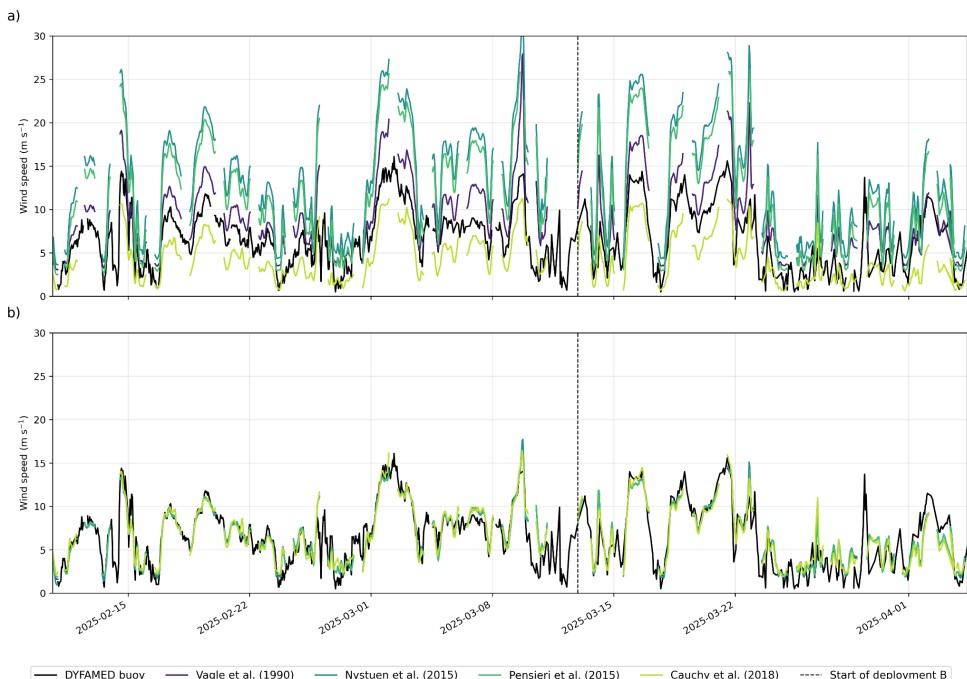

**Figure 4.** Comparison of unoptimized (top) and optimised (bottom) wind speed models against DYFAMED buoy observations. Each subplot shows modelled wind speed estimates from four literature models (Vagle et al., 1990; Nystuen et al., 2015; Pensieri et al., 2015; Cauchy et al., 2018) compared with collocated buoy wind data (black line). The unoptimized models a) use original published coefficients, while the optimised models b) are re-fitted using data within 40 km of the DYFAMED site. The dashed vertical line indicates the start of deployment B.

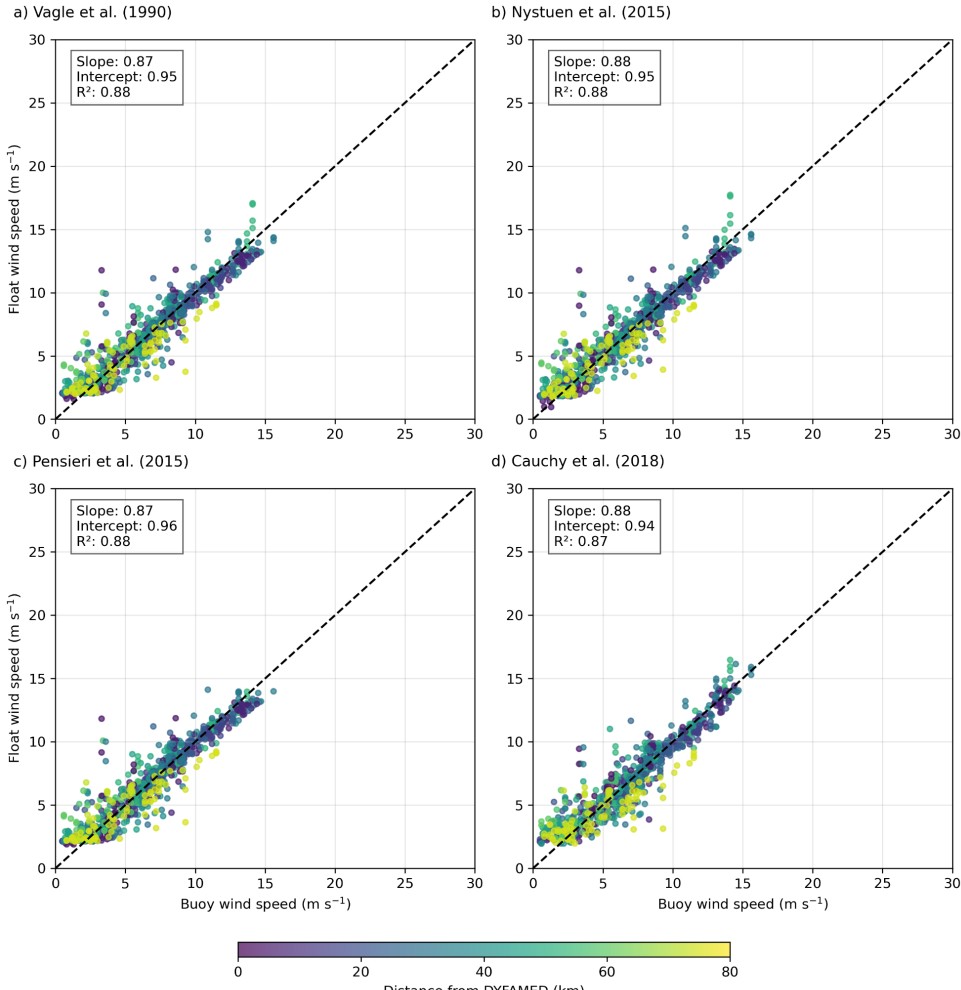

411

**Figure 5.** Comparison of optimised wind speed estimates from four literature models against collocated DYFAMED buoy wind measurements. Each subplot (a–d) shows scatter plots of float-derived wind speed vs. buoy wind speed using model-specific optimised coefficients: (a) Vagle et al. (1990), (b) Nystuen et al. (2015), (c) Pensieri et al. (2015), and (d) Cauchy et al. (2018). Points are color-coded by distance from the DYFAMED buoy, and the dashed line represents the 1:1 reference. Insets display linear regression slope, intercept, and coefficient of determination ($R^2$).






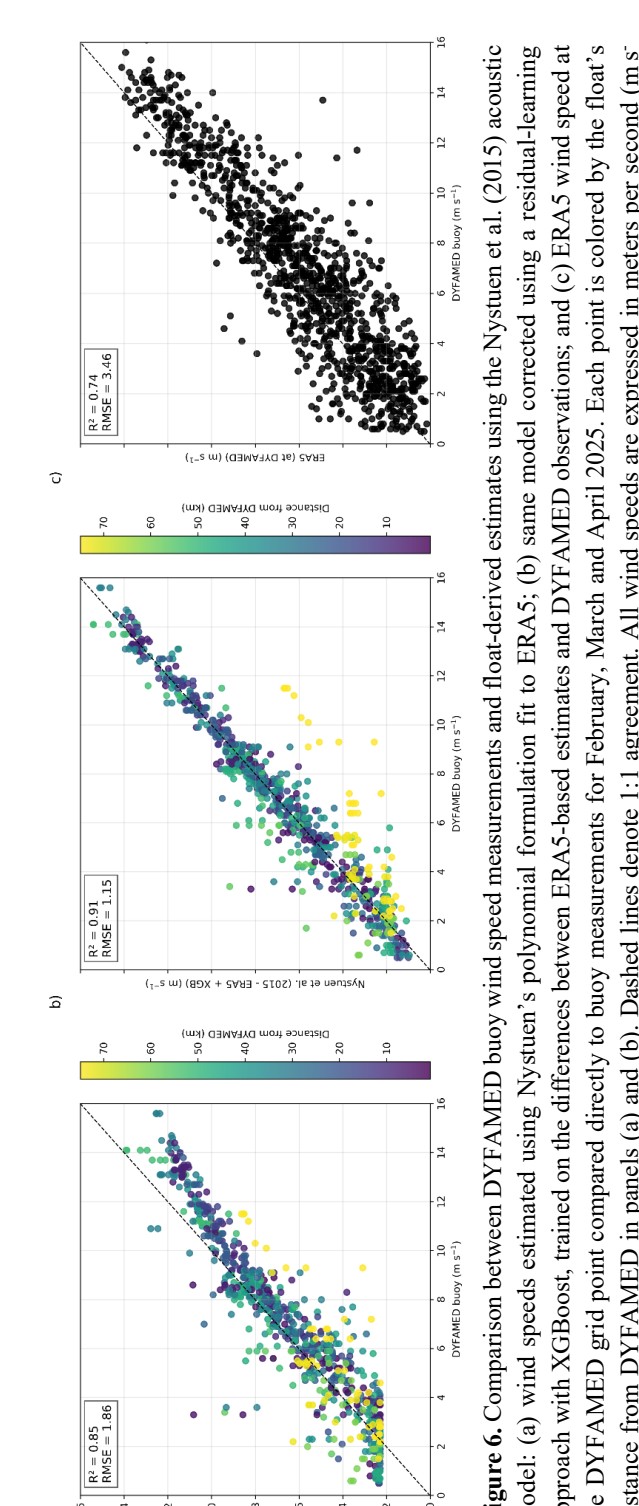

**Figure 6.** Comparison between DYFAMED buoy wind speed measurements and float-derived estimates using the Nystuen et al. (2015) acoustic model: (a) wind speeds estimated using Nystuen's polynomial formulation fit to ERA5; (b) same model corrected using a residual-learning approach with XGBoost, trained on the differences between ERA5-based estimates and DYFAMED observations; and (c) ERA5 wind speed at the DYFAMED grid point compared directly to buoy measurements for February, March and April 2025. Each point is colored by the float's distance from DYFAMED in panels (a) and (b). Dashed lines denote 1:1 agreement. All wind speeds are expressed in meters per second (m s$^{-1}$).






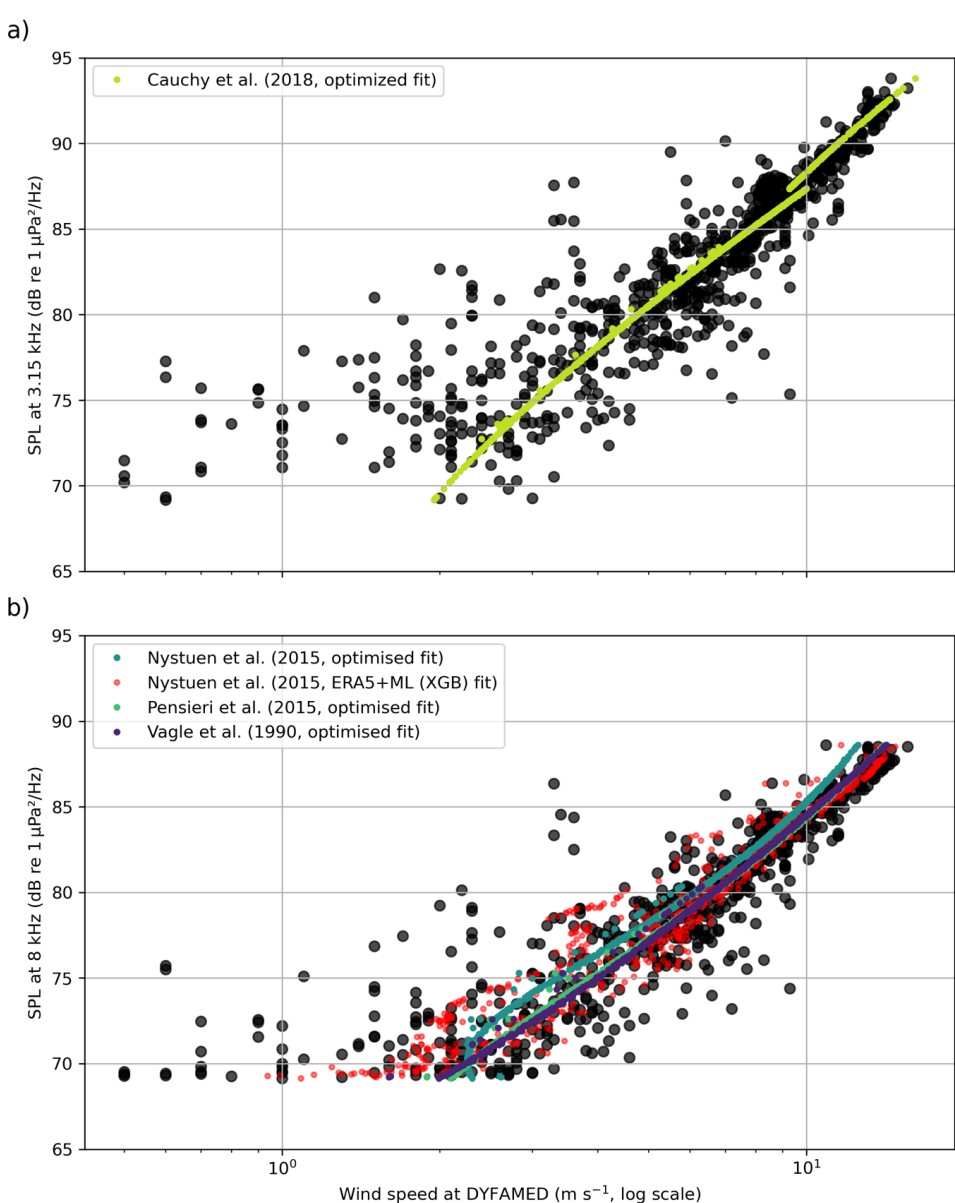


**Figure 7.** Optimised 10-meter wind speed (log scale) as a function of observed underwater
sound pressure level (SPL) at DYFAMED for (a) 3.15 kHz and (b) 8 kHz. Observed wind speed
is shown in black.


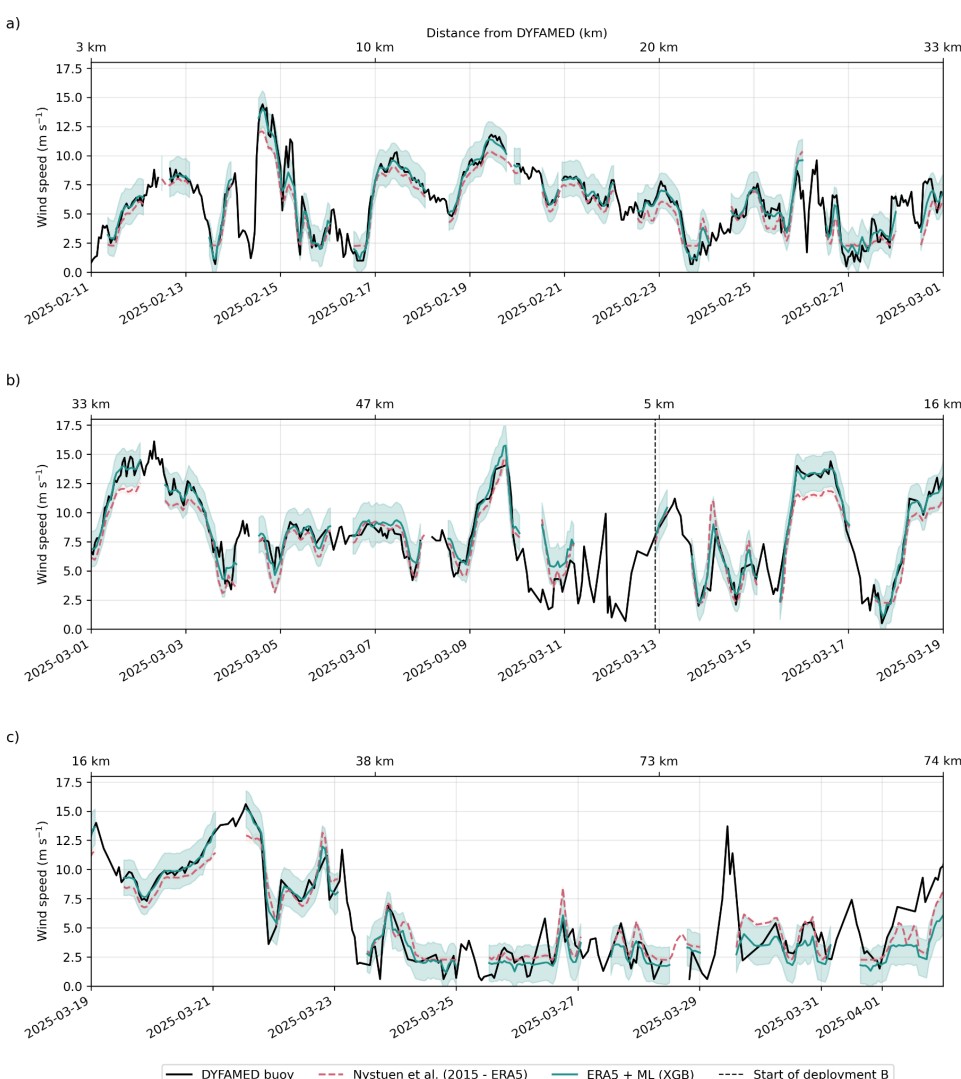

**Figure 8.** Time series comparison of wind speed estimates from the acoustic float and DYFAMED buoy observations, shown across three sequential 18-day segments of the deployment (a–c). The dashed pink line shows estimates from the Nystuen et al. (2015) model fit to ERA5-derived inputs. The solid green line represents the same model corrected using a residual-learning approach (XGBoost) with its associated uncertainty. Black curves show in-situ wind speed from the DYFAMED buoy. The top x-axis indicates the float's distance from DYFAMED over time, and a dashed vertical line marks the start of deployment B.





### 3.1 Assessing the performance of float-based acoustic wind estimation

We applied four previously published wind retrieval models to float-measured sound pressure
levels (SPLs) at 8 kHz and 3 kHz. Using the original coefficients from these studies, wind speed
estimates deviated significantly from collocated DYFAMED observations, particularly in their
ability to reproduce the magnitude of wind events (Fig. 4a). This mismatch reflects the
sensitivity of empirical acoustic models to deployment context, including platform geometry,
acoustic propagation, and local noise environment.
When these same models were refitted using collocated float acoustics and DYFAMED wind
observations within 40 km (Fig. 1), performance improved markedly (Fig. 4b; Fig. 7). Among
the models, the cubic formulation by Nystuen et al. (2015) achieved the best fit ($R^2 = 0.88$; Fig.
5b) and successfully captured the full observed wind range (0.5–16.1 m s$^{-1}$; Figs. 5 and 7).
Notably, it was the only model capable of resolving wind speeds below 2 m s$^{-1}$, a critical range
often underrepresented due to weak surface forcing and minimal bubble generation. This low-
end sensitivity is particularly valuable for air–sea gas exchange estimates in biogeochemical
studies and suggests that the Nystuen model may be more broadly applicable in low-to-
moderate wind regimes.
However, even after successful fitting, the transferability of acoustic–wind models remains
uncertain. Factors such as noise contamination, ambient biological activity and regional
propagation conditions can vary substantially between deployments, affecting both the shape
and robustness of the acoustic–wind relationship. Moreover, profiling floats introduce their
own artifacts, which may arise from hydrodynamic turbulence, buoyancy engine activity,
bubble release, or electronic interference, each of which can contaminate the acoustic signal
independently of wind forcing. In our study, even models originally developed in the same
basin required refitting (i.e. Pensieri et al. 2015; Figs. 4, 5 and 7), underlining the challenge of
cross-platform and cross-region generalization.
A promising future direction may involve grouping deployments into broader "acoustic
environment types"—such as open-ocean gyres, coastal shelves, or high-latitude storm
zones—within which shared model parameters could be defined and validated. This aligns with
the priorities outlined in the Ocean Sound Essential Ocean Variable (EOV) Implementation
Plan, which emphasizes the need for community-agreed metadata standards, calibration
protocols, and classification schemes to support global comparability across acoustic
deployments (Tyack et al., 2023). Evaluating the adequacy of such frameworks in the context
of profiling float–based wind retrieval could inform future updates and promote harmonization
with broader ocean observing efforts.

### 3.2 Generalizing float-specific wind modelling using reanalysis

While site-specific fitting of acoustic wind models yields accurate float-derived wind
estimates, such fittings are not feasible in most regions of the global ocean where in-situ wind
observations are unavailable. To assess whether the acoustic–wind relationship can be



generalized for remote deployments, we investigated the use of reanalysis wind products as a
proxy reference for model fitting. Specifically, we used the ERA5 atmospheric reanalysis (Bell
et al., 2021) to refit the Nystuen et al. (2015) model to float-measured acoustic data, simulating
a scenario where no collocated buoy or shipboard wind measurements are available (Figs. 6
and 8).
Using time-matched float sound pressure level at 8 kHz and collocated ERA5 wind speed, we
derived a new set of coefficients (Section 2.6), representing a general-purpose acoustic wind
model that could, in principle, be deployed globally using only float data and reanalysis inputs.
The objective of this exercise was not to develop a new region-specific model, but rather to test
whether existing models could be adapted—via reanalysis fitting—for use in data-sparse areas,
ultimately enabling scalable wind estimation from profiling floats globally.
As shown in Figure 6a, this ERA5-calibrated Nystuen et al. (2015) model reproduced wind
variability within the 2.5–10 m s$^{-1}$ range with moderate skill (R$^2$ = 0.85), and performed best
during Deployment A, when wind conditions remained relatively stable and within the
moderate wind regime (Fig. 8). However, performance declined during periods of stronger
wind, particularly in Deployment B (Figs. 6a and 8). In these cases, the model systematically
underestimated wind speeds, with errors exceeding 3 m s$^{-1}$ during high-wind events.
Comparison with ERA5 reanalysis also revealed broader limitations. Although ERA5 provides
a globally consistent reference product for surface winds, it diverged from DYFAMED data
during several high-wind episodes, especially in Deployment B. This discrepancy is consistent
with earlier studies reporting the underestimation of localized orographic wind events in
reanalysis datasets over semi-enclosed basins such as the Mediterranean (Bentamy et al., 2003;
Bell et al., 2021). This limitation is especially consequential for deployments in the Southern
Ocean, where high-wind regimes are frequent and drive a large share of the global air–sea $CO_2$
flux. Underestimating these events could lead to significant biases, as gas exchange scales
nonlinearly with wind speed (Wanninkhof, 2014; Wanninkhof et al., 2025).
Thus, while float-based acoustic wind estimation can be extended using reanalysis data in the
absence of in-situ observations, its accuracy ultimately depends on the fidelity of the reference
product used for fitting. In our case, reanalysis-based fitting performed well in moderate wind
regimes but failed to capture the intensity of high-wind events—highlighting the limitations of
relying solely on global reanalysis in dynamic or orographically complex regions.
**3.3 Simulating scalable wind estimation in data-sparse regions**
While reanalysis-calibrated acoustic models offer a pathway for estimating surface wind
speed in remote regions, the results in Section 3.2 show that this approach alone remains
insufficient during high-wind events or rapidly evolving conditions. This limitation poses a
significant challenge for air–sea interaction studies in the Southern Ocean and other high-





latitude regions, where extreme wind forcing drives critical fluxes of heat, momentum, and
carbon (Lee et al., 2017; Dotto et al., 2019; Zhang et al., 2022; Gruber et al., 2023).

### 511    3.3.1 Local model correction using residuals learning

To overcome this, we implemented a residual learning framework that combines the
generalizability of reanalysis-based fitting with the accuracy of localized corrections.
Specifically, we trained an ensemble of XGBoost regression models to predict the residuals
between the ERA5-calibrated estimates and collocated DYFAMED buoy observations (see
Section 2.6). The model was trained using float data within 40 km of DYFAMED and
bootstrapped over 100 iterations to estimate both mean corrections and predictive uncertainty
(Fig. 1; Fig. 6b). The 40 km radius was selected based on the sensitivity analysis of Cauchy et
al. (2018), who found it to balance proximity with data availability; however, this threshold
may be site-specific and should be re-evaluated in future deployments to reflect local acoustic
and meteorological conditions.
The corrected wind time series showed markedly improved alignment with DYFAMED
observations (Fig. 8), particularly during high-wind events where the uncorrected model
consistently underestimated wind speed. This bias-correction approach yielded a substantial
performance gain, increasing the coefficient of determination ($R^2$) from 0.85 to 0.91—an
absolute improvement of 0.06, or approximately 7.1% relative to the baseline model. At the
same time, the root mean square error (RMSE) dropped from $1.88 \, \text{m s}^{-1}$ to $1.15 \, \text{m s}^{-1}$,
corresponding to a 37.0% reduction in prediction error. While other learning-based methods
have achieved comparable improvements—e.g., Zambra et al. (2022) reported a 16% RMSE
reduction using a physics-informed deep learning model—our method differs by explicitly
using reanalysis as a prior and requiring only sparse in-situ fitting.
The machine learning model does not estimate wind speed directly. Instead, it learns to adjust
the bias based on a limited number of input features: acoustic signal intensity, deployment day,
and the ERA5-calibrated prediction. In essence, it identifies when and where ERA5 is likely to
fail, applying larger corrections under high-wind conditions where reanalysis tends to
underestimate variability.
The results demonstrate that even a limited number of in-situ fitting points—simulating, for
example, a brief engine-off ship-based wind measurement window during float deployment—
could significantly improve wind estimates across the full float trajectory. In our case, the in-
situ data used for fitting represented approximately 40% of the full dataset, due to the relatively
short deployment duration. However, this approach also introduces potential limitations. First,
although we aimed to simulate operational constraints, the fitting points were drawn from the
same dataset used for evaluation, raising the possibility of optimistic bias in the reported
performance. Future deployments should explore spatially or temporally distinct training–
validation splits or assess generalization using fully withheld reference stations. Second, the
observed reduction in RMSE reflects improvements primarily at the higher end of the wind





speed range, where raw model errors tend to be largest. While this benefits absolute RMSE
metrics, it may overstate improvements at lower wind speeds.

### 3.3.2 Strategies for sparse in-situ calibration

In practical terms, however, acquiring suitable reference observations can be challenging.
While ship-based wind measurements are a natural candidate—particularly during float
deployment or recovery—they may be unsuitable for model fitting if the ship is too close, as
engine noise can contaminate the float's acoustic signal. A viable compromise is to position
the ship nearby—but not too close—so that wind speed measurements remain representative
while minimizing acoustic interference. Alternatively, a more robust strategy is to deploy floats
in proximity to existing meteorological buoys, which provide collocated wind observations
without interfering with subsurface acoustic recordings.
In regions where neither buoys nor suitable ship data are available, identifying whether the
available in-situ coverage is sufficient becomes more complex. This will depend not only on
the duration and trajectory of the float mission, but also on the opportunistic use of additional
reference sources encountered along the way—for example, other buoys, or wind observations
from vessels transiting the area. In such cases, satellite-based products—particularly synthetic
aperture radar (SAR) imagery—could offer another valuable source of wind information.
These products provide high spatial resolution and can capture localized wind variability at
times and locations where in-situ data are sparse. Although episodic and weather-dependent,
SAR passes could serve as intermittent anchor points for model adjustment or evaluation.
More broadly, these scenarios highlight the need for flexible modelling approaches that can
exploit heterogeneous and temporally limited reference data. Rather than relying on dense
training datasets or persistent surface observations, future efforts could explore machine
learning paradigms such as domain adaptation, transfer learning, or few-shot learning, which
aim to adapt models to new environments with minimal retraining. For instance, recent work
by Wang et al. (2020) has shown that few-shot transfer methods can yield competitive
performance even when only a small number of target-domain samples are available.
In the context of profiling floats, such strategies could enable a more scalable approach to
acoustic model tuning, by leveraging sparse data from ships, buoys, or satellites—each with its
own limitations but collectively offering sufficient diversity. We propose framing this as
opportunistic multisource model fine-tuning: a hybrid calibration scheme in which local
corrections are derived from whatever reference sources are available, without requiring dense
or continuous in-situ coverage. Developing and validating such methods will be essential to
deploy acoustic-equipped floats globally while maintaining robustness across a wide range of
environmental and acoustic conditions.



### 3.3.3 Implications for global observing

While ERA5 provides a useful climatological reference, it tends to underestimate short-lived, high-wind events due to spatial and temporal smoothing. This is an issue particularly for gas exchange studies, as extreme winds disproportionately contribute to total fluxes. Acoustic float data—collected continuously and at high resolution—are uniquely positioned to detect these events, even when they fall below the detection threshold of satellite or reanalysis products.

However, model performance degrades with increasing distance from DYFAMED, reflecting the spatial decorrelation of wind fields and the limited spatial representativeness of the buoy observations. Beyond 73 km during Deployment B, both the Nystuen et al. (2015) – ERA5 fit and the machine-learning-corrected float estimates begin to diverge from DYFAMED winds (Figs. 6 and Fig. 8). This divergence does not necessarily imply model failure but rather raises the possibility that the float and buoy are sampling different wind regimes. In such cases, it becomes difficult to determine whether discrepancies are due to limitations in the acoustic model or to true spatial variability in wind forcing. One way to address this uncertainty is to analyse float trajectories that pass between two surface reference stations, assessing whether refitting at the final station yields consistent corrections or reveals systematic regional shifts in wind decorrelation. Such an approach will require future deployments that span multiple buoys, enabling a systematic evaluation of how model performance degrades—or remains robust— across both time and space.

Additionally, in the Southern Ocean, where anthropogenic noise is relatively low, it may also be worth reconsidering the use of lower-frequency bands (<1 kHz) for wind estimation. These frequencies are more sensitive to high wind speeds due to increased bubble activity and longer propagation ranges and may outperform higher-frequency bands under strong forcing conditions—provided contamination from distant shipping or other sources remains minimal.

Several recent studies have applied machine learning to underwater acoustic data to estimate wind and rainfall, often relying on long-term, stationary deployments and direct prediction from spectral features (Taylor et al., 2020; Trucco et al., 2022; Trucco et al., 2023; Zambra et al., 2022). While these approaches have shown strong performance under controlled conditions—such as Taylor et al.'s use of moored PAL systems during storm events or Zambra et al.'s assimilation-based deep learning scheme—they typically require dense, labelled datasets and assume relatively stable acoustic environments.

In contrast, our residual learning strategy is designed for sparse, mobile deployments. It corrects reanalysis-based estimates using short-duration in-situ fitting and does not require full acoustic training labels, making it more adaptable to the practical constraints of autonomous profiling floats. While in-situ data remains the most difficult to obtain in remote, data-poor regions, our approach is well-suited to opportunistic fitting—for instance, using brief ship-based wind observations during deployment or leveraging nearby meteorological buoys. This hybrid strategy balances scalability with realism, enabling robust performance even in hard-to-access areas where long-term reference data are limited or unavailable.



In parallel, another important consideration is the potential for regional bias introduced by the
depth correction applied to acoustic levels. This correction compensates for propagation losses
due to local water column properties (e.g., temperature, salinity, and sound speed) and is
typically derived from the float's hydrographic profile at the start of the deployment. When
used to adjust the full acoustic time series, this introduces a location-dependent correction that
may vary across floats or missions. Ideally, the correction should be recalculated for each new
hydrographic profile, especially in long-term or wide-ranging deployments where temperature
and salinity conditions evolve. To ensure comparability of wind estimates at basin or global
scales, such corrections should be clearly documented and incorporated into standard
processing protocols for acoustic-equipped floats.
This deployment-focused flexibility is key to scaling up acoustic wind estimation globally. By
leveraging reanalysis products for first order fitting and applying localized corrections when
available, our framework enables accurate, event-resolving wind estimates without the need
for long-term surface infrastructure. Scaling this approach across the BGC-Argo array would
provide high-resolution, all-weather wind monitoring in regions poorly served by existing
networks.
**4 Conclusions**
This study provides the first demonstration of retrieving surface wind speeds from subsurface
ambient noise recorded by a profiling float equipped with a passive acoustic sensor. By
integrating a low-power hydrophone onto an autonomous profiling float and applying
established acoustic retrieval algorithms, we successfully detected surface wind variability
from depths between 500 and 1000 m. When empirically calibrated using collocated buoy
observations, float-derived wind speed estimates closely matched in-situ surface
measurements, confirming the feasibility and accuracy of this approach under realistic
oceanographic conditions.
To evaluate its potential for application in remote, data-sparse regions, we simulated a scenario
where acoustic models were calibrated solely using ERA5 reanalysis winds. Although the
ERA5-based calibration captured moderate wind variability effectively ($2.5$–$10\,\mathrm{m\,s^{-1}}$), it
consistently underestimated high-wind events, underscoring limitations in using reanalysis data
as a standalone reference. To mitigate this, we implemented a residual-learning approach,
leveraging brief periods of local wind observations (e.g., from ship-based or moored
instruments) to correct systematic errors in the acoustic estimates. This hybrid methodology
substantially improved model performance, particularly under high-wind conditions,
maintaining accuracy across extended float trajectories and demonstrating robustness for
operational use.
These findings underscore the potential of acoustic-equipped profiling floats as scalable and
autonomous platforms capable of delivering high-resolution surface wind observations in
remote or poorly instrumented oceanic regions. Such observations are particularly critical for



refining estimates of air–sea exchanges, including the oceanic uptake and release of $CO_2$,
processes significantly influenced by wind-driven gas exchange. Combined with emerging
biogeochemical proxy algorithms, such as CANYON-B and CONTENT, acoustic-equipped
floats can now provide fully autonomous, integrated estimates of air–sea $CO_2$ fluxes by
coupling accurate wind measurements with concurrent measurements of oceanic temperature,
salinity, and oxygen.
Nevertheless, this study represents a single deployment in a semi-enclosed basin. Broader
validation across diverse oceanographic regimes, including open-ocean gyres, polar regions,
and high-energy storm zones, is necessary to fully assess the robustness, generalizability, and
temporal stability of the proposed correction frameworks. Future deployments will help refine
the methods presented here and further test their applicability across different acoustic
environments and platform configurations.
The demonstrated capability to retrieve accurate wind speeds from subsurface acoustic
measurements marks a significant advancement in autonomous ocean observing. As next-
generation passive acoustic sensors become increasingly integrated into the global BGC-Argo
array, this technology offers a cost-effective and efficient strategy for addressing persistent
observational gaps. Such developments will enable unprecedented insights into wind forcing,
air–sea interactions, and climate-relevant ocean processes in regions historically challenging
to monitor through traditional methods.
Looking forward, the ability to calibrate acoustic wind retrievals using sparse local reference
measurements not only improves float-based wind estimates but also provides a valuable new
data stream for validating and potentially correcting biases in global wind reanalyses. As
acoustic-equipped floats accumulate data across various ocean regions, their observations may
substantially enhance the fidelity of global atmospheric products, particularly in remote areas
currently lacking validation data.
Ultimately, this work aligns closely with the Ocean Sound Essential Ocean Variable (EOV)
Implementation Plan, advocating for standardized methodologies, robust metadata
documentation, and interoperable frameworks across acoustic observing platforms.
Demonstrating successful acoustic wind retrieval from autonomous, mobile platforms thus
contributes directly to the practical realization of global observing standards, strengthening the
integration of passive acoustics into sustained, multidisciplinary ocean observing systems.



**Funding.** The research leading to these results has received part of the funding from the European Union's Horizon research and innovation program under grants #101094716 (GEORGE project) and #101188028 (TRICUSO project). REFINE has received funding from the European Research Council (ERC) under the European Union's Horizon 2020 research and innovation programme (grant agreement N° 834177). Argo-2030 has received the support of the French government within the framework of the "Investissements d'avenir" program integrated in France 2030 and managed by the Agence Nationale de la Recherche (ANR) under the reference "ANR-21-ESRE-0019".

**Data availability.** The two deployments of this prototype float have not been assigned a WMO identifier and have not been declared in Argo; the data are therefore not available through the Argo program. All float data, DYFAMED buoy measurements, ERA5 reanalysis wind fields, and analysis scripts used in this study is freely available online at https://doi.org/10.5281/zenodo.17232551.The repository include processed datasets, code for model fitting and residual learning, and figure-generation scripts to ensure full reproducibility of results.

**Author contributions.** EL, HC, and LD conceptualized the project. AD and CS developed the acoustic sensor used in this study. LD curated the data. EL, HC, and LD performed the investigation. LD conceptualized the methodology, used the necessary software, visualized the data, and prepared the original draft of the paper. AGM, DC, EL, HC, JB, LD, PC, RB and SP reviewed and edited the paper.

**Competing interests.** NKE instrumentation is a private company which commercialized the acoustic float, in which AD and CS are employed. The acoustic float is based on the PROVOR CTS5 platform and on an acoustic sensor developed and commercialized by NKE instrumentation with a partnership agreement with LOV. All other co-authors declare no competing interests.

**Disclaimer.** Publisher's note: Copernicus Publications remains neutral with regard to jurisdictional claims in published maps and institutional affiliations.

**Acknowledgements.** We gratefully acknowledge the DYFAMED team at the Laboratoire d'Océanographie de Villefranche for their support in deploying the float, recovering it, and enabling its second deployment. We also thank the crew of the Téthys for their assistance at sea, and Jean-Yves for facilitating two successful float recoveries on short notice. We are grateful to the European Centre for Medium-Range Weather Forecasts (ECMWF) for making the ERA5 reanalysis products freely available, which provided an essential reference dataset for this study. We also thank Aldo Napoli (Mines Paris) for his assistance with the AIS data and Ambroise Renaud (Mines Paris) for making his modified version of *libais* freely available on GitHub, which allowed us to parse NMEA data from the AIS reception device at Mines Paris (Sophia Antipolis) and from the AISHub feed. We are also grateful to AISHub (AIS data sharing and vessel tracking by AISHub) for providing access to their AIS data services.



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
