# Peer review of "Passive acoustic monitoring from profiling floats as a pathway to scalable autonomous observations of global surface wind"

_EGUsphere, 2025_

## Author Comment (AC1)

**RC1: 'Comment on egusphere-2025-4174', Anonymous Referee #1, 07 Nov 2025**

The manuscript represents an outstanding contribution to scientific progress by demonstrating a scalable, autonomous system for wind speed estimation using profiling floats and passive acoustic monitoring (PAM).

The core novelty is the successful deployment and retrieval of wind data from deep parking depths (500–1000 m) and the development of a residual learning framework. This framework is critical because it addresses the core limitation of acoustic wind retrieval in remote regions: the lack of local calibration data. By combining global reanalysis (ERA5) with sparse in-situ observations to correct systematic biases, the authors achieved a major quantitative improvement: a 37% reduction in RMSE and an increase in $R^2$ from 0.85 to 0.91. This work provides a practical path to integrate wind forcing observations with the BGC-Argo float array.

The scientific approach and applied methods are valid and well-justified. The use of established empirical models (like Nystuen et al., 2015) and the innovative application of the XGBoost algorithm for residual learning are appropriate for handling the non-linear relationship between acoustics and wind. The necessary preprocessing steps, such as depth correction and noise mitigation, are included.

The discussion is appropriate and balanced, explicitly acknowledging the systematic bias of the ERA5-fitted model in high-wind regimes (>10 m/s) and the need for the residual correction. It accurately situates the findings within the framework of prior moored and mobile PAM research.

The authors present the scientific results and conclusions in a clear, concise, and well-structured manner. The manuscript adheres to a standard, logical flow, and the technical language is precise. The figures are of high quality and clearly show the main findings, especially when comparing the unoptimized, ERA5-fitted, and ML-corrected time series (Figure 8) and the scatter plots (Figure 6 Tables clearly present all necessary methodological details, including frequency band integration times and model requirements.

Given the groundbreaking nature of the results and the high overall quality, the manuscript should be accepted subject to minor revisions.

We thank the reviewer for their very positive and constructive assessment of the manuscript.

The "minor revisions" category is suggested to meet the need for a clearer explanation of the current validation strategy. Suggested minor revisions focus on enhancing the discussion (Section 3.3.1, paragraph 5) and the conclusions by explicitly stating that while the framework works, the performance metrics may represent the upper bound of expected accuracy due to using the same short-duration deployment for both training and validation.

Following the reviewer's suggestion, we have revised Section 3.3.1 (paragraph 5) and the Conclusions to explicitly acknowledge this limitation and to clarify that the reported metrics likely represent an upper bound:

Section 3.3.1 – added sentence: "Taken together, these factors imply that the reported performance metrics likely represent an upper bound of the framework's accuracy for long-duration or multi-region deployments."

Conclusions – added sentence: "Nevertheless, our results stem from a single short-duration deployment. Broader validation across regions, seasons, and acoustic environments is needed, and performance estimates likely represent an upper bound. Recent benchmarking efforts (e.g., Gros-Martial et al., 2025) already demonstrate the value of assembling multi-site acoustic–meteorological datasets and highlight the challenges of model transferability across diverse soundscapes."

Expand the need for future work to validate the model's generalizability using spatially or temporally distinct training-validation splits to confirm the framework's robustness for global, remote deployment.

L691-693 (Conclusions) – added sentence: "Future missions should employ independent training–validation–test partitions to rigorously evaluate generalizability, following best practices established in recent WOTAN studies that explicitly address temporal correlation and multi-site validation requirements (e.g., Cauchy et al., 2018; Taylor et al., 2020; Trucco et al., 2022; Trucco et al., 2023)."

---

## Author Comment (AC2)

**RC2: 'Comment on egusphere-2025-4174', Anonymous Referee #2, 17 Nov 2025**

The manuscript presents a novel deployment of a profiling float equipped with a passive acoustic sensor for estimating surface wind speed from subsurface ambient noise. The work is timely and relevant, especially for the Ocean Sound EOV and emerging multisensor BGC-Argo platforms. The authors provide an extensive evaluation of several established acoustic wind models, propose a combined reanalysis–residual correction framework, and assess the performance of their approach using a deployment near the DYFAMED site. Overall, the paper is well written and contains substantial technical detail. The results indicate promising capability for autonomous wind sensing from profiling floats. However, several aspects require clarification, tightening, or additional evidence before the paper can be recommended for publication. I outline the main points below.

We thank the reviewer for their careful reading of the manuscript and for the constructive comments. We address the reviewer's points below and have revised the manuscript accordingly to improve clarity, strengthen the methodological justification, and better support the robustness of the results.

Major issues

1. Length and focus of the Introduction: The Introduction covers too many tangential topics and becomes diffuse. Several paragraphs repeat similar background points (e.g., wind relevance, WOTAN history, BGC-Argo capabilities). The core motivation of the study - why wind sensing on modern profiling floats matters and what gap is being addressed - would be clearer with a more concise and focused introduction. Some reduction would improve readability.

We substantially revised and streamlined the Introduction to improve clarity, focus, and narrative cohesion. The updated version is now ≈400 words shorter than in the original submission. Beyond simply shortening the text, the revised Introduction now provides a more direct and integrated framing of the study. It opens by emphasising the persistent observational gaps in wind measurements across remote ocean regions, where existing satellite and in situ platforms face well-known limitations. It then introduces passive acoustic monitoring as a mature yet underutilised approach for retrieving wind speed from subsurface platforms, summarising key advances across moorings, drifters, gliders, and biologging studies. The new version also highlights the expanding role of passive acoustics in geophysical and ecological observing, aligning our work with broader community priorities such as the Ocean Sound Essential Ocean Variable (EOV). Importantly, the streamlined text clarifies the specific gap our study addresses: the absence of demonstrated wind retrievals from the deep parking depths used by modern BGC-Argo floats, and the lack of a practical framework for integrating acoustic observations with reanalysis products to enable

scalable wind estimation in data-sparse regions. Together, these revisions produce a more concise, coherent, and motivation-driven Introduction that better guides the reader toward the study's objectives and scientific contributions.

2. Novelty requires clearer articulation: Previous studies (Riser et al., Yang et al., Ma et al.) have demonstrated wind estimation using float-mounted acoustic sensors, though with more limited onboard processing and telemetry. The manuscript mentions this but does not explicitly define what is distinct about the system used here and what the scientific advance is. The paper would benefit from a short statement clearly outlining the new elements (e.g., full third-octave spectra transmission, integration into CTS5 BGC-Argo, post-processing flexibility, residual-learning framework).

The last paragraph of the introduction was rewritten as follows:

"In this study, **we present the first deployment of a biogeochemical profiling float equipped with a passive acoustic sensor explicitly designed for wind speed estimation from subsurface ambient noise**. Deployed in the northwestern Mediterranean Sea, near the DYFAMED (DYnamique des Flux Atmosphériques en MEDiterranée) meteorological buoy, **this float serves as a proof-of-concept demonstration to: (1) determine whether wind-driven acoustic signatures can be detected at profiling float parking depths; (2) evaluate the performance of established acoustic wind models on this platform; and (3) develop a practical framework combining acoustic observations with reanalysis data to enable wind estimation in remote regions.** Through this approach, we demonstrate the potential of acoustic-equipped profiling floats to expand global wind observations, close persistent observational gaps, and support interpretation of biogeochemical and climate-relevant processes."

3. Methodological thresholds need justification: Several choices (40 km DYFAMED radius, 99th percentile transient filter, 3-hour smoothing window, 10 km AIS radius + RMSE anomaly rule) appear somewhat arbitrary. Some are based on previous work, but the conditions differ enough that sensitivity analysis is warranted. I suggest authors should explain why these specific thresholds were chosen and demonstrate that the results are not overly sensitive to them. For example, 40 km spatial filter - justified by Cauchy et al. (2018), but the conditions differ (glider vs float, different months, bathymetry). 99th percentile transient noise removal - is this threshold too aggressive? Could it remove real high-wind events? The 10 km AIS radius and RMSE-based anomaly filter is somewhat circular, because RMSE is computed relative to the same reference (DYFAMED) used for filtering.

We thank the reviewer for raising this important point. Several methodological thresholds in the original manuscript were indeed insufficiently justified. In the revised version, we now offer a better justification for each choice.

We thank the reviewer for this comment. The 40 km radius used here is not dependent on platform type (glider vs float), deployment month, or bathymetry. In Cauchy et al. (2018), this threshold was derived from the spatial decorrelation scale of the wind field itself, evaluated in the same NW Mediterranean region using DYFAMED winds. The decorrelation length therefore reflects a regional atmospheric property rather than a platform-specific constraint. Because our deployment took place in the same basin, under similar meteorological regimes, the 40 km scale remains directly applicable. We have clarified this point in Section 2.6 and now explicitly state that this radius represents a regional mesoscale decorrelation length rather than a glider-specific parameter. While this threshold is appropriate for the NW Mediterranean, we agree that it should be re-evaluated for other regions or seasons in future missions.

Section 2.6 now includes: "Although originally derived from the spatial wind-field decorrelation scale reported by Cauchy et al. (2018), this 40 km radius reflects a regional mesoscale atmospheric property rather than a platform-specific constraint. Because our deployment occurred in the same NW Mediterranean basin, this decorrelation length remains appropriate for our case. We note, however, that this threshold is region-dependent and should be re-evaluated for future deployments elsewhere."

For the transient filters, we have clarified Section 2.5 which now reads:

"2.5 Transient and anthropogenic noise mitigation

Transient noise (i.e., episodic non-wind-related events) was mitigated by removing values exceeding the 99th percentile within a ±1.5-hour window centred around each matched timestamp. This percentile corresponds to discarding roughly the top 1% of samples over a 3-hour window—about two minutes of data. No physically meaningful wind- or wave-driven variability relevant to this study evolves on such short timescales, making this filter effective at removing brief acoustic artefacts without suppressing real high-wind conditions. This approach is conceptually similar to the transient-noise mitigation used in glider-based PAM studies (e.g., Cauchy et al., 2018), which suppress short-lived spikes in the spectra to isolate wind-generated noise.

To further reduce short-term variability and emphasize quasi-stationary wind-driven acoustic patterns, we applied a 3-hour rolling mean to each frequency band. This smoothing window is conceptually consistent with the profile-scale averaging used in glider-based acoustic wind studies (e.g., Cauchy et al., 2018), where acoustic measurements are aggregated over ~2-hour glider dives to suppress transient variability. While smoothing inevitably attenuates rapid fluctuations, the 3 h window stabilises the spectra without erasing multi-hour wind events relevant for air–sea flux

applications. Alternative strategies, such as post-processing the wind speed estimates rather than the spectral bands, could be explored in future deployments if finer-scale variability is a priority.

Anthropogenic noise was mitigated using AIS vessel tracks. Because the float only provides GPS positions at the surface, we reconstructed a continuous trajectory by linearly interpolating its positions between successive surfacings at hourly resolution. Each 5-min acoustic record was then associated with the nearest interpolated position. An observation was flagged as potentially contaminated when an AIS-reported vessel was located within 20 km of this interpolated float position and within ±30 min of the acoustic timestamp. The 20 km radius corresponds to the distance over which ship-radiated noise commonly dominates the ambient sound field in the 1–10 kHz band under low-to-moderate sea states, while the ±30 min window accounts for the typically irregular AIS reporting interval offshore. As an additional safeguard, we excluded cases where the float-derived wind speed deviated from the DYFAMED buoy by more than the RMSE computed under uncontaminated conditions. This RMSE criterion is used only as a secondary check to capture possible contamination during periods of poor AIS coverage. Sensitivity tests indicate that moderate changes to these thresholds do not affect the main conclusions."

Please consider additional sensitivity tests, or clearer justification acknowledging limitations.

While a full sensitivity analysis is beyond the scope of the present deployment, the revised Methods now state more clearly that these thresholds were chosen to follow established practice, ensure comparability with prior acoustic studies, and retain sufficient data volume. We also emphasise in the Discussion that future long-duration or multi-float deployments should reassess these thresholds and explore formal sensitivity analyses.

4. Depth-correction assumptions: The $\beta(h,f)$ correction is mathematically complex. However, it is applied only once from the first CTD profile, yet temperature–salinity changed over ~60 days. Authors state conditions were "relatively stable," but this should be shown quantitatively ($\Delta T$, $\Delta S$, $\Delta c$ sound speed). Please provide evidence that using only one correction introduces < X dB error. I suggest adding a small analysis showing that variability in T/S over the deployment would not meaningfully change $\beta$, or revise the text to state this is a limitation.

In response to the reviewer's request to quantify hydrographic stability over the deployment, we assessed the temporal variability of the water-column structure using profiles that reached at least 1000 dbar and included sufficient near-surface sampling. Each cast was interpolated onto a 1-m grid between 0 and 1000 dbar, and anomalies were computed relative to the deployment-mean temperature and salinity profiles. Depth-averaged RMS deviations were 0.14 ± 0.04 °C for temperature and

0.06 ± 0.02 for salinity, with no cast exceeding two standard deviations from the mean (|z| < 2 for all profiles). In other words, none of the deep profiles spanning the entire deployment differed significantly from the mean hydrographic structure.

To evaluate the impact of this hydrographic variability on the acoustic depth-correction, we compared β(h,f) computed from the first CTD profile with β(h,f) recomputed using the deployment-mean profile. The maximum absolute difference in Δβ(h,f) was 0.014 dB, the 95th-percentile difference was 0.013 dB, and the RMS difference was 0.008 dB across the 20–1000 m depth range at both 3.15 kHz and 8 kHz. Because these values are negligible relative to instrumental variability and the acoustic dynamic range used for wind retrieval, recomputing β(h,f) for every cast is unnecessary. Nevertheless, for internal consistency, the analysis presented here now uses the β(h,f) derived from the mean temperature–salinity profile.

The first paragraph of section "2.3 Depth correction and spectral normalization" now reads:

"To account for the attenuation of surface-generated noise with depth, a correction term β(h,f) was applied to all acoustic measurements (Fig. 2). Because β depends on the ambient temperature–salinity structure, we quantified hydrographic stability over the 60-day deployment using all profiles that reached at least 1000 dbar. Each profile was interpolated onto a 1 m grid and compared to the deployment-mean temperature/salinity profiles. Depth-averaged RMS deviations were 0.14 ± 0.04 °C for temperature and 0.06 ± 0.02 for salinity, and no profile exceeded |z| = 2 standardized deviation, confirming weak hydrographic variability. Because such differences are far below hydrophone measurement uncertainties, β(h,f) was computed once using the deployment-mean profile and applied uniformly to the full record. For longer or more dynamic missions, β(h,f) should be recomputed for each profile. Modern hardware makes this operation computationally inexpensive, but the negligible hydrographic variability in this deployment renders repeated recalculation unnecessary."

5.  ERA5-based model fitting and validation: While the ERA5-calibrated model performs reasonably in moderate winds, the validation relies entirely on the same buoy used later for residual correction. This risks circularity. The conclusion that the method resolves high-frequency wind variability "not captured by ERA5" is plausible but not demonstrated quantitatively. A more rigorous validation strategy (e.g., cross-validation, leave-one-event-out) would strengthen the claims.

We acknowledge that the ERA5-calibrated model and the residual-learning correction are both evaluated against the DYFAMED buoy, which introduces the possibility of optimistic skill estimates. As noted in our response to Reviewer 1, we have now made this limitation explicit in both Section 3.3.1 and the Conclusions. In particular,  we clarify that the reported performance represents an upper bound

because training and validation occur on the same short-duration deployment, and that a rigorous assessment of generalizability will require spatially or temporally distinct validation datasets.

At present, the duration and spatial extent of the available dataset do not permit a rigorous cross-validation or leave-one-event-out scheme without severely under-sampling the training set. For this reason, the ERA5-based calibration is presented primarily as a feasibility demonstration rather than a full validation exercise. To acknowledge this clearly, we have amended the end of Section 3.3.1 as follows: "Taken together, these factors imply that these performance metrics likely represent an upper bound of the framework's accuracy for long-duration or multi-region deployments. The generalisation across sites, seasons and events remains untested and will require validation using spatially or temporally independent datasets.'"

We also revised a sentence in the abstract for consistency with this clarification: the original statement "This framework enables the retrieval of fine-scale wind variability not captured by reanalysis alone" was changed to "This framework enhances agreement with in-situ wind observations relative to reanalysis alone". The end of the introduction also now mentions the fact that this study is a proof-of-concept.

Finally, to address the reviewer's concern about over-stated claims regarding the ability of our method to resolve high-frequency wind variability, we revised several sentences in Section 3.3.3 to adopt more cautious and evidence-aligned wording.

Specifically, the original sentence "Acoustic float data—collected continuously and at high resolution—are uniquely positioned to detect these events, even when they fall below the detection threshold of satellite or reanalysis products" was replaced with "Acoustic float data, collected continuously and at high resolution, offer the potential to complement satellite or reanalysis wind products, particularly during short-lived wind events that are smoothed out in coarse-resolution products.

Finally, the statement "our framework enables accurate, event-resolving wind estimates without long-term surface infrastructure" was softened to "our framework improves agreement with in-situ winds without requiring long-term surface infrastructure."

Together, these changes ensure that the manuscript accurately reflects the level of validation achieved while avoiding claims that cannot be demonstrated quantitatively with the present dataset.

6. The residual learning section requires more detail: The machine-learning component is incompletely described. Important aspects (feature selection, hyperparameters, training sample size, prevention of overfitting) are missing.

Even if the intention is not to emphasize the ML model itself, some transparency is needed to assess whether the improvements are robust.

We agree that additional transparency was needed. In the revised manuscript, we have fully restructured Section 2.7, which is now split into two subsections (2.7.1 ERA5-based calibration of the acoustic model and 2.7.2 Residual -learning correction using limited in-situ observations). Subsection 2.7.2 has been substantially expanded to provide a clear description of the machine-learning component, including the feature set, training sample size (40 km collocations), model choice (XGBoost), hyperparameters, and the measures used to limit overfitting (bootstrap resampling, shallow trees, subsampling, and Gaussian perturbations of ERA5 inputs). We also now describe the 100-member ensemble used to characterise uncertainty. These additions provide the transparency required to assess the robustness of the residual-learning correction

The last paragraph of Section 2.7.2 Residual -learning correction using limited in-situ observations also offers more ML details:

"Residuals between DYFAMED wind speed and the ERA5-calibrated acoustic estimate were modelled using four predictors: SPL at 8 kHz, ERA5 10-m wind speed, normalised deployment day, and the Nystuen-model wind estimate. These variables capture the local acoustic signal, large-scale atmospheric forcing, slow temporal drift, and the first-order empirical fit. Residuals were estimated with XGBoost regression (Chen & Guestrin, 2016), using all float–buoy collocations within 40 km (~40% of the dataset). To maintain generalisation, we applied a compact hyperparameter set (300 estimators, learning rate 0.05, max depth 3, subsample 0.9, colsample_bytree 0.8) together with safeguards against overfitting, including bootstrap resampling, Gaussian perturbations of ERA5 winds ($\sigma = 1.5$ m s$^{-1}$) during training and prediction, shallow trees, and subsampling of both rows and features. Uncertainty was quantified using a 100-member ensemble, with each model trained on a bootstrap resample of the DYFAMED-matched subset and forced with perturbed ERA5 winds. This dual bootstrapping captures variability associated with the machine learning model structure and with ERA5 uncertainty. Corrected wind speeds were obtained by adding the ensemble-mean residual to the ensemble-mean Nystuen estimate, with total uncertainty expressed as $\pm 1\sigma$ by combining the XGBoost ensemble spread and ERA5 input uncertainty in quadrature. The bootstrap uncertainty of the Nystuen fit is reported separately. This framework provides a transparent and robust correction method, illustrating how float acoustics, reanalysis winds, and sparse in-situ observations can be combined to estimate surface wind speed in remote regions."

7. Discussion and Results are blended; consider restructuring: Much of Section 3 reads as discussion rather than results, particularly, sections 3.2 and 3.3 read more like discussion. The Results section should first present the findings

objectively and interpretation should follow separately. I suggest moving more speculative content (e.g., "future deployments," "few-shot learning") to Discussion.

We thank the reviewer for this observation. We agree that clear separation between results and interpretation is important. After revisiting Section 3, we concluded that our current structure follows the conventions typically used in observational oceanography and environmental acoustics, where methodological evaluation and data-driven interpretation are presented together to allow the reader to assess performance in context. In particular, Sections 3.2 and 3.3 describe the behaviour of the different acoustic models and the residual-learning framework directly alongside their quantitative evaluation, which we feel is essential for conveying the practical implications of the results.

That said, we have carefully reviewed the text and made targeted adjustments to ensure that interpretative statements are clearly distinguished from descriptive results. We hope the reviewer agrees that these refinements improve clarity while preserving the logical coherence of the results.

Minor Issues

- The abstract could be tightened; several sentences repeat key points.

The abstract has been revised.

- Fig. 8 is dense and difficult to interpret at first reading.

While we retained the overall structure of Fig. 8, which is necessary to illustrate both temporal evolution and distance to DYFAMED, we improved its readability by refining the caption, ensuring consistent colour contrasts and line weights across all panels, and renaming the plotted curves to make their meaning immediately clear to the reader.

- Some notations vary across equations (e.g., TOL vs SPL).

TOL and SPL refer to two different, sequential quantities (band-integrated third-octave levels vs. bandwidth-normalised spectral density). We have clarified this in Sect. 2.3 by explicitly defining SPL(f) as the depth-corrected, bandwidth-normalised quantity derived from $TOL_0(f)$ and stating that this convention is used in all subsequent equations:

"Then, depth-corrected third-octave levels $TOL_0(f)$ (in dB re 1 μPa) were converted to spectral density levels SPL(f) (in dB re 1 μPa²/Hz) by normalising to the bandwidth of each band. In the following, SPL always refers to these depth-corrected, bandwidth-normalised values derived from $TOL_0(f)$."

- A few acronyms are not defined at first appearance. (e.g., DYFAMED defined at line 141 but appeared several times prior).

DYFAMED is now defined directly in the abstract and again at its first occurrence in the Introduction.

- Several long sentences could be shortened for clarity.

We have reviewed the manuscript and shortened numerous long or complex sentences throughout to improve clarity and readability while preserving the scientific meaning. We have also substantially revised and condensed the Conclusions section, improving its structure and flow to provide a clearer, more concise synthesis of the study's key findings and implications.

- Line 164 & 167 third-octave → one-third octave?

We thank the reviewer for the suggestion. We chose to retain third-octave because this is the terminology used consistently in the ocean-acoustics literature (e.g., Nystuen et al., 2015; Baumgartner et al., 2017) and in the IEC standard for underwater acoustic measurements (IEC 61260-1). The meaning is identical to one-third-octave, and third-octave is the conventional form in this field. Therefore, no change was made.